# Identification of *Glaesserella parasuis* and Differentiation of Its 15 Serovars Using High-Resolution Melting Assays

**DOI:** 10.3390/pathogens11070752

**Published:** 2022-07-01

**Authors:** Simone Scherrer, Fenja Rademacher, Roger Stephan, Sophie Peterhans

**Affiliations:** Section of Veterinary Bacteriology, Institute for Food Safety and Hygiene, Vetsuisse Faculty, University of Zurich, CH-8057 Zurich, Switzerland; fenja.rademacher@uzh.ch (F.R.); stephanr@fsafety.uzh.ch (R.S.); sophie.peterhans@uzh.ch (S.P.)

**Keywords:** high-resolution melting, *G. parasuis*, molecular typing, high-resolution melting, serovar, virulence

## Abstract

*Glaesserella parasuis* is the etiological agent of Glässer’s disease, which is associated with polyserositis and arthritis and has a significant impact on the economy of the pig production industry. For the optimal surveillance of this pathogen, as well as for the investigation of *G. parasuis*-associated diseases, it is crucial to identify *G. parasuis* at the serovar level. In this work, we designed and developed new high-resolution melting (HRM) approaches, namely, the species-specific GPS-HRM1 and two serovar-specific HRM assays (GPS-HRM2 and GPS-HRM3), and evaluated the sensitivity and specificity of the assays. The HRM assays demonstrated good sensitivity, with 12.5 fg–1.25 pg of input DNA for GPS-HRM1 and 125 fg–12.5 pg for GPS-HRM2 and GPS-HRM3, as well as a specificity of 100% for the identification of all recognized 15 *G. parasuis* serovars. Eighteen clinical isolates obtained between 2014 and 2022 in Switzerland were tested by applying the developed HRM assays, which revealed a heterogeneous distribution of serovars 2, 7, 4, 13, 1, and 14. The combination with virulence marker *vtaA* (virulence-associated trimeric autotransporters) allows for the prediction of potentially virulent strains. The assays are simple to execute and enable a reliable low-cost approach, thereby refining currently available diagnostic tools.

## 1. Introduction

After a detailed phylogenetic analysis, *Haemophilus parasuis* was renamed *Glaesserella* (*G*.) *parasuis* [1], which is a globally emerging pig pathogen and etiological agent of Glässer’s disease that has a significant impact on the economy of swine production [2]. *G. parasuis* is a Gram-negative bacterium belonging to the *Pasteurellaceae* family, and it can be found in many commercial swine farms as a colonizer of the nasal cavity of piglets, where it can be recovered two days post-natal [3]. On the one hand, *G. parasuis* can act as a porcine commensal bacterium of the upper respiratory tract with different degrees of virulence [4,5]; on the other hand, especially in weaner pigs, some virulent strains can lead to serious systemic diseases, such as polyserositis, polyarthritis, and meningitis [6,7]. The onset of Glässer’s disease generally occurs in piglets between 1 and 4 months of age due to the stress of weaning or relocation in combination with a decrease in maternal immunity [8,9]. Another factor that favors disease onset is the co-infection of *G. parasuis* and viruses, such as the highly pathogenic porcine reproductive and respiratory syndrome virus (PRRSV) and/or swine influenza virus [10,11]. Not only is *G. parasuis* responsible for Glässer’s disease, but it is also implicated in the complex of swine respiratory disease [12], either as a predisposing factor as a secondary pathogen, as a primary pathogen for bronchopneumonia [7,13], or as the causal agent of acute septicemia [14]. The clinical signs of acute Glässer’s disease comprise typically swollen joints with lameness, a high fever above 41.5 °C, abdominal breathing, trembling, paddling, and coughing [15]. Pigs with moderate clinical signs can progress to a chronic stage with a reduced growth rate and lameness, and they typically show rough hair [7]. Affected farms present a mortality rate associated with Glässer’s disease between 5 and 10% [9]. Several strains of *G. parasuis* can colonize piglets at once, and, thus, in a single farm, multiple strains of *G. parasuis* can be found [3]. Initially, it was thought that serovars would be an essential virulence marker [16]. However, various research groups have reported that no strict correlation between serovars and virulence can be observed [17,18,19,20,21].

There is a broad genetic variability within the *G. parasuis* species mediated through plasmids and/or transposons [22]. A phylogenic analysis reported the existence of two main lineages, namely, lineages I and II, revealing an association with serovars, countries, and multilocus sequence types. A *G. parasuis* strain from Switzerland was identified to be part of lineage I, which mainly comprises serovars 5, 12, and 14. Lineage II predominantly includes serovars 2 and 10 [22]. Recently published studies about the distribution of *G. parasuis* in North America, Europe, and Asia found that the geographic prevalence of molecular serovars varied slightly among different countries; however, due to the variable sizes of the examined samples, no differences in serovar prevalence could be determined [23]. Overall, serovars 1, 4, 5/12, and 7 were most often identified, whereas serovars 4 and 5/12 were widely distributed across all regions included in the study [23].

For disease control, it is important to identify the correct serovar and the virulence potential of isolates in order to differentiate between commensals and virulence-associated *G. parasuis* strains. In 2015, Howell et al. [24] published a multiplex serotyping PCR, which was developed using serovar-specific primers based on the capsule loci capable of differentiating 14 of the 15 known *G. parasuis* serovars without discrimination between serovars 5 and 12. Jia et al. [25] developed a multiplex PCR, which allows for differentiation between all recognized 15 serovars; however, this assay uses different annealing temperatures, causing the identification of the correct serovar to be time and work intensive. Besides these serotyping assays, other PCR assays have been developed to trace the virulence markers of *G. parasuis*. A group of specific virulence-associated genes, called virulence-associated trimeric autotransporters (*vtaA*) group 1-translocator, has been found to be related to virulent *G. parasuis* strains [26]. Furthermore, in *vtaA* genes, two different leader sequences have been identified, and an assay differentiating between putative virulent and non-virulent *G. parasuis* strains based on the leader sequence of the *vtaA* gene, called LS-PCR, has been developed [5].

It would be greatly advantageous to have a molecular tool that combines the identification of *G. parasuis,* determining its putative virulence, and, at the same time, having an accurate discrimination between all existing serovars. A high-resolution melting (HRM) analysis is a low-cost and rapid method based on PCR, and it examines the dissociation behavior of PCR amplicons. An incremental increase in temperature after PCR completion leads to the dissociation of double-stranded DNA into single-stranded molecules, thus resulting in a decline in fluorescence intensity. The melting temperature (Tm) for each amplicon is specific and depends on the amplicon sequence, its GC content, and its length [27].

In this work, a new high-resolution melting (HRM) approach was developed combining the identification, serotyping, and prediction of the putative virulence of *G. parasuis* isolates as a simple and cost-efficient molecular assay.

## 2. Results

### 2.1. HRM Assay

The designed primers, as indicated in Table 1, Table 2 and Table 3, were divided into two assays (GPS-HRM2 and GPS-HRM3) considering their distinctive Tm to obtain the best possible resolution power in order to enable the differentiation between all 15 serovars. GPS-HRM1 allows for the simultaneous identification of *G. parasuis*, using HPS_219690793 as a species-specific target and *vtaA* for differentiation between virulent and non-virulent *G. parasuis* isolates. All 15 *G. parasuis* reference strains amplified the gene HPS_219690793, and the subsequent HRM showed a melting curve with a Tm of 75.8 ± 0.2 for serovars 1, 2, 4, 5, 7, 11, 12, 13, 14, and 15, and a Tm of 75.7 ± 0.2 for serovars 3, 6, 8, 9, and 10, clearly identifying *G. parasuis*. Additionally, the non-virulent *G. parasuis* strains of serovars 3, 6, 8, 9, and 10 successfully amplified *vtaA* using primers AV1-F and NV1-R, represented by a melting curve with a Tm of 79.65 ± 0.3. Therefore, the group of putative virulent strains could clearly be differentiated from the non-virulent isolates, obtaining either one melting peak for the virulent group or two peaks for the non-virulent group (Figure 1).

All identified *G. parasuis* isolates obtained using GPS-HRM1 were further analyzed by simultaneously applying two different primer mixes (GPS-HRM2 and GPS-HRM3) in parallel. Combining the results of GPS-HRM2 and GPS-HRM3 enabled the differentiation of all 15 serovars of *G. parasuis* as a consequence of distinct melting curves for each serovar (Figure 2). For each serovar, a distinct Tm and its standard deviation are listed (Table 2 and Table 3). Generally, the estimated resolution power of GPS-HRM1 was 0.3 centigrade, whereas for both HRM assays GPS-HRM2/-HRM3, a difference of 0.15 centigrade was resolvable.

The coefficient of variation (CV) of the variability of the assays obtained when using GPS-HRM1 to target HPS_219690793 resulted in CV% ≤ 0.07% for intra-assay variability and CV% ≤ 0.06% for inter-assay variability. The second target of GPS-HRM1 (*vtaA*) yielded CV% ≤ 0.05% for intra-assay variability and CV% ≤ 0.03% for inter-assay variability (Appendix A). GPS-HRM2 and GPS-HRM3 revealed CV% ≤ 0.09% for intra-assay variability and CV% ≤ 0.14% for inter-assay variability, respectively. The low CV values obtained for each GPS-HRM assay suggest high reproducibility and robustness (Appendix A).

### 2.2. Analytical Specificity

Analyzing the 27 different bacterial strains with the use of GPS-HRM1 for exclusivity testing did not give rise to any melting curves, and, thus, the negative results reveal a specificity of 100%. Moreover, all 15 *G. parasuis* reference strains did not show any cross-reactions among the different serovars, and this is illustrated as unique melting curves for each serovar (Figure 2).

### 2.3. Analytical Sensitivity

Standard curves were acquired by testing the tenfold dilution series using the genomic DNA of the 15 *G. parasuis* reference strains when performing GPS-HRM1, which amplified the *G. parasuis*-specific gene locus HPS_21969079 and *vtaA* associated with non-virulent isolates (Appendix A). GPS-HRM2 and GPS-HRM3 were performed by amplifying 15 different primer pairs specific to each serovar, which resulted in 15 different standard curves (Appendix A). The linear range of the standard curves obtained with GPS-HRM1 was between 50 and 5 × 10^6^ GE for the 15 *G. parasuis* reference strains. At a confidence level of 95%, the following LODs were identified: 5 GE (12.5 fg) for *G. parasuis* serovar 15; 50 GE (125 fg) for *G. parasuis* serovars 1, 2, 5, 6, 7, 8, 9, 10, 11, 12, 13, and 14; and 500 GE (1.25 pg) for *G. parasuis* serovars 3 and 4. GPS-HRM1 revealed high correlation coefficients of R^2^ > 0.99 for the standard curves. In Appendix A, an example of a representative standard dilution series of *G. parasuis* illustrates the high sensitivity of *G. parasuis* when performing GPS-HRM1. The standard curve showed linearity across a broad range of DNA concentrations ranging from 5 to 5,000,000 GE.

GPS-HRM2 and GPS-HRM3 had a more variable linear range of standard curves than GPS-HRM1. LODs above the relevant confidence level of 95% were found to be 50 GE (125 fg of genomic DNA) for *G. parasuis* serovars 1, 2, and 3; 500 GE (1.25 pg of genomic DNA) for *G. parasuis* serovars 4, 5, 6, 7, 8, 9, 10, and 15; and 5000 GE (12.5 pg of genomic DNA) for *G. parasuis* serovars 11, 12, 13, and 14. The correlation coefficients of the standard curves were R^2^ > 0.95 (Appendix A).

GPS-HRM1 had a LOD between 5 and 500 genome equivalents (GE) (12.5 fg–1.25 pg) of DNA and PCR efficiencies between 96 and 105%. GPS-HRM2 and GPS-HRM3 had LODs between 50 and 5000 GE (125 fg–12.5 pg) genomic DNA and PCR efficiencies between 92 and 105%.

### 2.4. Efficiency

The efficiency of GPS-HRM1 was between 96 and 107% for each of the 15 *G. parasuis* reference strains. The PCR efficiencies of GPS-HRM2 and GPS-HRM3 were between 92 and 105% (Appendix A).

### 2.5. Clinical Isolates

The Tm values obtained from the HRM assays testing the eighteen clinical isolates were compared to the Tm values obtained from the 15 *G. parasuis* reference strains. The eighteen clinical isolates were recognized to be *G. parasuis* by the species-specific primers. No HRM signal was obtained for the eighteen clinical isolates with the PCR when using primers detecting non-virulent *G. parasuis* isolates, thus revealing that the analyzed isolates were virulent. Using the HRM assays GPS-HRM2 and GPS-HRM3, the correct serovars of the eighteen isolates were unambiguously uniquely identified by referring to the Tm values (Figure 3). In total, 38.9% (*n* = 7) of the isolates were found to be *G. parasuis* serovar 2, 22.2% (*n* = 4) were found to be serovar 7, 16.7% (*n* = 3) were found to be serovar 4, 11.1% (*n* = 2) were found to be serovar 13, and 5.6% (*n* = 1) was found for both serovars 1 and 14.

## 3. Discussion

Due to the severe clinical signs caused by *G. parasuis*, it is essential to fight the pathogen appropriately. Besides strict hygienic measures, antimicrobial treatments and vaccines are widely used tools to combat Glässer’s disease [29]. The knowledge about globally increasing antimicrobial resistance underlines the importance of vaccination for the prevention of *G. parasuis* infection. The successful development of vaccines targeting certain virulent and prevalent strains requires a reliable and fast identification of predominant *G. parasuis* serovars in affected farms.

With the development of the novel HRM approaches, performing GPS-HRM1 allows for the efficient identification of *G. parasuis* and its potential virulence while performing GPS-HRM2 and GPS-HRM3 leads to the differentiation of *G. parasuis* serovars. The assays demonstrated a specificity of 100% for all 15 *G. parasuis* serovars, since none of the 15 reference strains resulted in non-specific signals. Using GPS-HRM1 to target HSP_219690793, *G. parasuis* showed a LOD of 5-500 GE (12.5–1250 fg) of input DNA, illustrating good sensitivity. In comparison to a previously standardized real-time PCR assay targeting *infB*, the parameters, such as the correlation coefficient and efficiency values, were in the same range [28]. The serotyping assays GPS-HRM2 and GPS-HRM3 only reached a detection limit of 50–5000 GE (0.125–12.5 pg) of input DNA and, therefore, did not achieve the same sensitivity as GPS-HRM1 because of the higher complexity of the PCR master mix, which comprised up to eight primer pairs.

Performing qPCR followed by an HRM analysis is a very convenient closed-tube procedure requiring few manipulation steps. In contrast to a conventional PCR, which includes the analysis of PCR products by capillary or agarose gel electrophoresis, the novel HRM assays can be executed more efficiently in a laboratory with no downstream processing of samples. Therefore, performing HRM is less time consuming, and, at the same time, the cost of resources is reduced. Additionally, data can easily be accessed and interpreted in comparison to the recognition of sometimes-difficult band patterns obtained with a conventional PCR. Considering the sensitivity of the HRM assays, genomic DNA between 12.5 fg and 12.5 pg was sufficient to obtain a result; this in contrast to a conventional PCR, which requires more genomic template DNA (typically in the nanogram range) to obtain a signal, further underlining the advantages of the developed assay.

It is important to mention that primers targeting the *funB* of *G. parasuis* serovar 1 cross-react with the genomic DNA of serovar 11; thus, serovars 1 and 11 result in the same melting peaks when using GPS-HRM2 (Figure 2A). However, an unambiguous identification can be acquired by performing GPS-HRM3 using serovar-11-specific primers targeting *amtA.* Combining the results of GPS-HRM2 and GPS-HMR3 allows for the unique assignment of serovar 11 and serovar 1, respectively (Figure 2B). Notably, new serovars can potentially be identified when obtaining a *G. parasuis* species-specific HRM signal using GPS-HRM1 in combination with the absence of a serovar-specific melting curve in GPS-HRM2 or GPS-HRM3.

For diagnostic purposes, the serotyping of *G. parasuis* is time consuming and needs a lot of resources. Consequently, in many routine diagnostic laboratories, serotyping is often not performed. However, it is of the outmost importance to identify the virulence and serovars of involved strains for the correct interpretation of their relevance. With the knowledge of implicated serovars, autogenous vaccines can be created, and disease spread can be eradicated. Furthermore, the novel HRM assays can help to overcome the extensive use of antibiotics. In the future, it is planned to evaluate this method for application with DNA directly extracted from tissue material or swabs.

As shown in the exclusivity study, the HRM assays allow for the differentiation of *G. parasuis* strains from their most important differential agents, such as *Escherichia coli*, *Streptococcus suis*, *Mycoplasma hyorhinis*, and *Mycoplasma hyosynoviae*.

The vaccines available in Switzerland are based on serovar 5 (Porcilis Glässer, MSD Animal Health GmbH, Lucerne, Switzerland). Furthermore, commercially available bacterin vaccines in other countries contain a combination of serovars 4 and 5 or serovars 1 and 6 [9]. However, cross-protection against other serovars is often reduced [30]. Since different serovars are found in one farm, or even in one animal, it would be favorable to identify and type relevant systemic isolates and to use those as an autogenous vaccine [9]. Therefore, these novel assays can help to reliably identify the appropriate serovar and can help to maximize suitable protection.

In Switzerland, serovar identification of the eighteen clinical *G. parasuis* isolates gathered between 2014 and 2022 illustrated a heterogenous distribution of serovars 2, 7, 4, 13, 1, and 14 in declining order at frequencies of 38.9–5.6% each (Table 4). However, no statement about the real prevalence can be made since the small number of eighteen isolates is not meaningful. It would be interesting to monitor the continuance of *G. parasuis* in the future in order to verify the observation that serovar 5/12, unlike in most other countries, is absent in Switzerland and, eventually, to determine the reason for the particular serovar distributions of *G. parasuis*.

## 4. Materials and Methods

### 4.1. Bacterial Strains and Clinical Isolates

For the development of novel HRM assays, the following *G. parasuis* reference strains of serovars 1–15 were examined: nr.4, SW140, SW114, SW124, Nagasaki, 13l, 174, C5, D74, H555, H465, H425, 84-17975, 84-22113, and 84-15995 (Table 5).

Eighteen clinical isolates were collected between 2007 and 2022 from routine diagnostic submissions to the Section of Veterinary Bacteriology, University of Zurich (Table 4), and they were analyzed with the novel HRM assays. The non-virulent strains were either not among the samples obtained from diagnostic submissions or could not successfully be isolated.

Clinical samples were grown on solid chocolate agar (Thermo Fisher Diagnostics AG, Pratteln, Switzerland) for up to 48 h at 37 °C in 5% CO_2_.

### 4.2. HRM Conditions

Primers were designed with the sequences of *G. parasuis* retrieved from the NCBI databank, using gene loci based on previous reports, or, alternatively, primers were directly adapted from previous publications [5,24,28]. Primers targeting sequences HSP-219690793, *glyC* (serovar 3 specific), and *wcwK* (serovar 12 specific) were designed by taking into consideration the optimal distribution of the different Tms of the peaks. Briefly, the nucleotide sequences of the primers were shifted and sized within the target region in order to fit HRM criteria, such as optimizing the amplicon length to smaller than 160 basepairs. Primer sequences were selected for the assay whenever the interval of the Tms of the different serovars was clearly distinctive. The specificity testing of the primer sequences was performed using a BLAST analysis. All primers were manufactured by Microsynth (Balgach, Switzerland). The HRM assays were developed using a Rotor-Gene Q (Qiagen, Hilden, Germany). The extraction of genomic DNA was performed using InstaGene Matrix (Bio-Rad Laboratories, Richmond, VA, USA) according to the manufacturer’s instructions. DNA concentration was determined using spectrophotometry with a NanoDrop 2000c (Thermo Fisher Scientific, Waltham, MA, USA).

The PCR reaction was performed in a total reaction volume of 15 µL. The reaction mix included 1 µL of genomic DNA, 7.5 µL of Type-it HRM PCR Master Mix (2×) (Qiagen), the primers at the concentrations listed (Table 1, Table 2 and Table 3), and water. The following PCR thermocycling parameters were applied: denaturation at 95 °C for 5 min, 40 cycles at 95 °C for 10 s, annealing/extension either at 57.5 °C (GPS-HRM1) or at 61 °C (GPS-HRM 2 and GPS-HRM3) for 30 s; and a final cycle at 95 °C for 10 s and at 40 °C for 2 min. HRM ramping was performed from 62 °C to 95 °C, acquiring data of the fluorescence at 0.1 °C increments every 2 s in order to create melting plots specific to each serovar. As a positive control for the PCR, genomic DNA extracted from the reference strains was used in each HRM run. Distilled water was applied as a negative control in each HRM assay.

Using 15 reference strains (Table 5), the HRM assays were developed. The melting curves were analyzed using Rotor-Gene Q Software 2.3.1 (Qiagen). Using this software, a melt curve analysis was performed. The Tm of the obtained peaks was examined using a threshold of 0.5 dF/dT. Signals below the threshold of 0.5 dF/dT were not considered.

To assess the repeatability of GPS-HRM1, GPS-HRM2, and GPS-HRM3, 20 ng of genomic DNA of all 15 serovar reference strains was examined in triplicate, thereby evaluating the intra- and inter-assay variabilities of the obtained Tm of each HRM assay.

### 4.3. Analytical Specificity

An exclusivity panel of 27 bacterial isolates encompassing eleven different species was examined with the three assays GPS-HRM1, GPS-HRM2, and GPS-HRM3 to evaluate analytical specificity. The following strains comprising three groups were tested: Gram-negative coccobacilli, namely, *Actinobacillus suis* (*n* = 1), *Pasteurella multocida* (*n* = 3), *Actinobacillus pleuropneumoniae* (*n* = 3), and *Bordetella bronchiseptica* (*n* = 2); nasal commensal microorganisms, such as *G. parasuis*: *Moraxella* spp. (*n* = 1) and *Neisseria animaloris* (*n* = 1); and bacteria involved in causing lesions similar to those caused in Glässer’s disease, namely, *Streptococcus suis* (*n* = 5), *Erysipelothrix rhusiopathiae* (*n* = 1), *Escherichia coli* (*n* = 2), *Mycoplasma hyorhinis* (*n* = 4), and *Mycoplasma hyosynoviae* (*n* = 4).

### 4.4. Analytical Sensitivity

For the evaluation of the analytical sensitivity of the three HRM assays, the 15 G. parasuis reference strains were analyzed. With an estimation of the genome size of G. parasuis between 1.4 and 2.2 Mbp [19], one genome equivalent (GE) corresponded to approximately 2.5 fg of genomic DNA.

To determine the detection range and linearity of GPS-HRM1, GPS-HRM2, and GPS-HRM3 using the 15 *G. parasuis* reference stains, a triplicate testing of a tenfold standard dilution series was performed comprising genomic DNA of 12.5 ng (5 × 10^6^ GE), 1.25 ng (5 × 10^5^ GE), 125 pg (5 × 10^4^ GE), 12.5 pg (5 × 10^3^ GE), 1.25 pg (500 GE), 125 fg (50 GE), and 12.5 fg (5 GE). The correlation coefficient (R^2^) represents the linearity of each standard dilution series of the 15 *G. parasuis* serovars. For the determination of the limit of detection (LOD) in a confidence interval of 95%, the lowest concentration of each serovar was chosen, at which a melting curve for all triplicates with a threshold value greater than 0.5 dF/dT and a standard deviation of the crossing threshold (Ct) greater than 0.5 was obtained.

### 4.5. Efficiency

To calculate the PCR efficiency in the linear range of the dilution series, the following equation was used: efficiency = (10^1/-slope^ − 1) × 100. In order to calculate the slope of the standard curves, C_t_ values obtained from triplicate measurements of the dilution series (5 × 10^6^ GE, 5 × 10^5^ GE, 5 × 10^4^ GE, 5 × 10^3^ GE, 500 GE, 50 GE, and 5 GE) of the three HRM assays tested were plotted against GE. Using the formula mentioned above, the efficiency of each *G. parasuis* serovar was determined.

### 4.6. Clinical Isolates

Eighteen clinical isolates were further characterized as described in Section 4.1 by applying primer mixes for GPS-HRM1, GPS-HMR2, and GPS-HRM3. For the assignment of the serovars, the Tm received from each clinical isolate was compared with the corresponding Tm obtained from the 15 *G. parasuis* serovar reference strains.

## 5. Conclusion

The novel HRM assays, which address, at the same time, the identification of *G. parasuis* and its potential virulence (GPS-HRM1), as well as differentiation between all 15 serovars (GPS-HRM2 and GPS-HRM3), deliver a practical diagnostic method that can provide rapid insight into possible pathogenic stains. The assays can be used in diagnostic laboratories to obtain a broad picture about the prevalence of *G. parasuis* among pig farms, thereby opening new perspectives among the currently available diagnostic methods.

## Figures and Tables

**Figure 1 pathogens-11-00752-f001:**
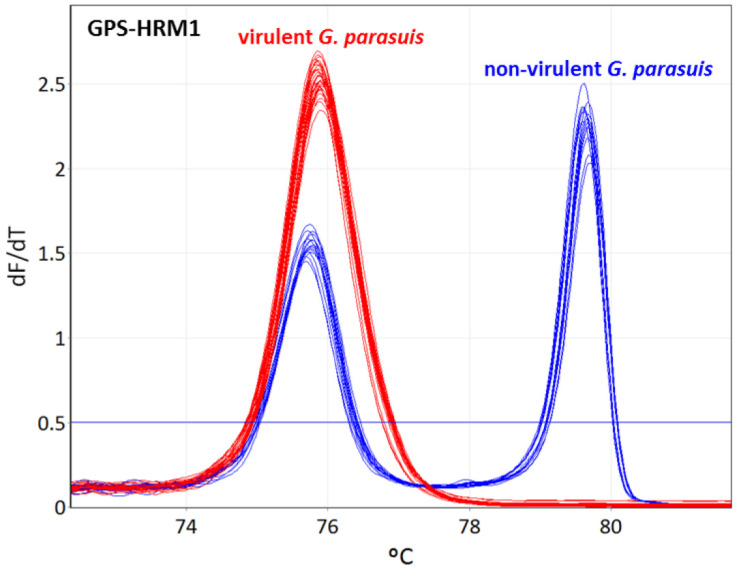
Identification of *Glaesserella parasuis* and differentiation between virulent and non-virulent strains using high-resolution melting. GPS-HRM1 assay amplifies the species-specific gene HPS_219690793, identifying *G. parasuis* and *vtaA* for discrimination of virulent and non-virulent *G. parasuis* strains, respectively. *G. parasuis* serovars 1, 2, 4, 5, 7, 11, 12, 13, 14, and 15 are illustrated in red (virulent strains), whereas serovars 3, 6, 8, 9, and 10 are illustrated in blue (non-virulent strains).

**Figure 2 pathogens-11-00752-f002:**
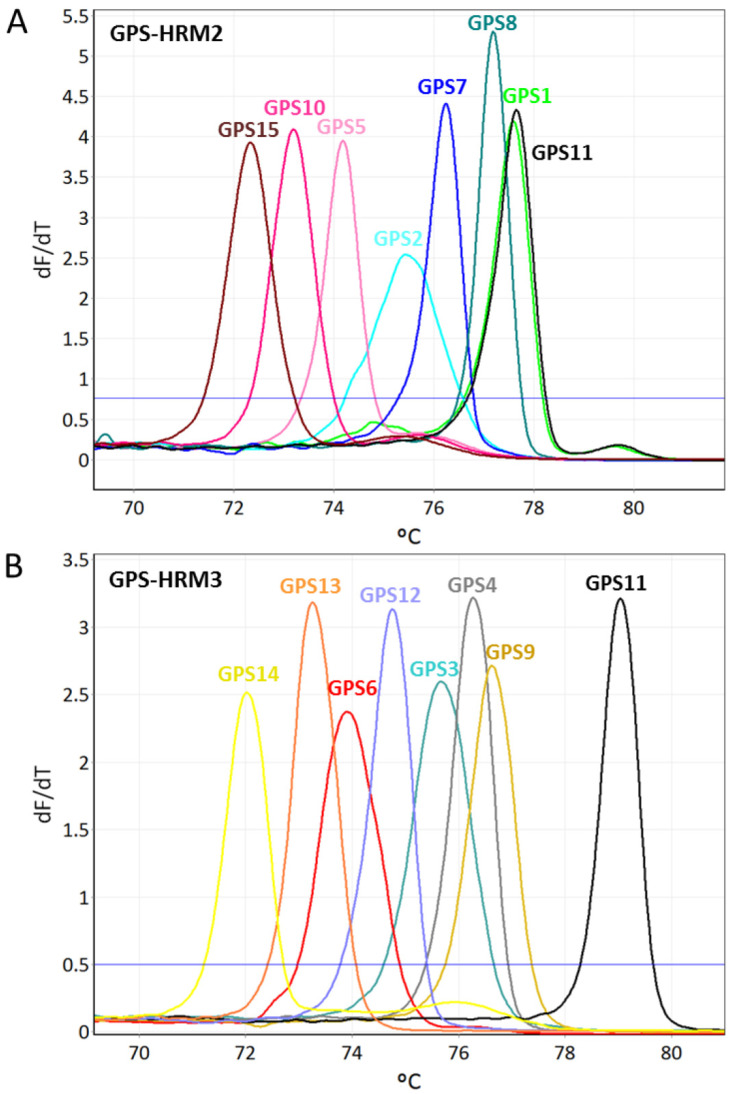
Illustrations of high-resolution melting (HRM) assays GPS-HRM2 (**A**) and GPS-HMR3 (**B**), capable of identifying all recognized 15 serovars of *Glaesserella parasuis.* Using GPS-HRM2, six serovars can be differentiated, as well as one group consisting of *G. parasuis* serovar 1/11. GPS-HRM3 allows for differentiation of eight serovars and, thus, resolves the grouping of serovar 1/11 by separately assigning serovar 11 and uniquely identifying serovar 1.

**Figure 3 pathogens-11-00752-f003:**
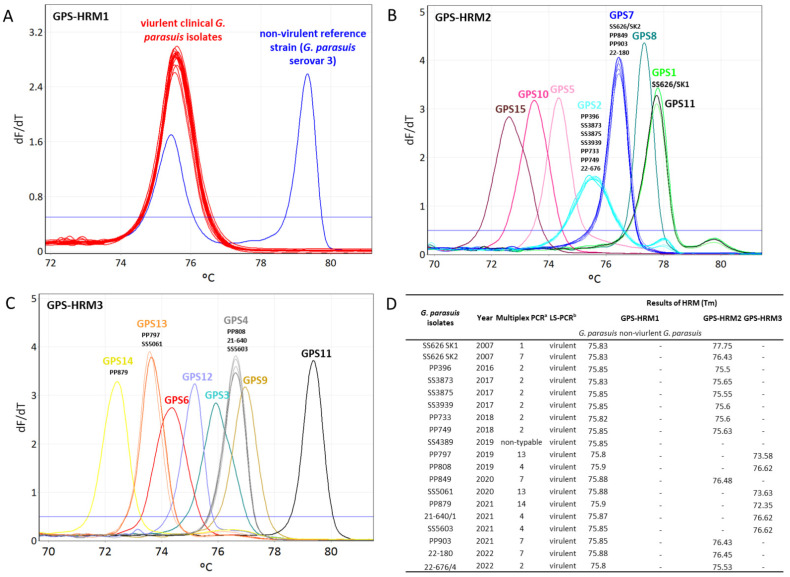
Serotyping using a high-resolution melting (HRM) analysis of eighteen Swiss clinical *Glaesserella parasuis* isolates dating from 2007 to 2022. Representation of HRM melting curves using (**A**) GPS-HRM1, (**B**) GPS-HRM2, and (**C**) GPS-HRM3. (**D**) Results of melting temperature (Tm) acquired from GPS-HRM1, GPS-HRM2, and GPS-HRM3. GPS-HRM1 was performed using *G. parasuis* serovar 3 (SW114) as a positive control; in GPS-HRM2 and GPS-HRM3, all 15 *G. parasuis* reference strains were used as positive controls. ^a^ serovar determination by multiplex PCR [24]; ^b^ virulence prediction by leader sequence PCR [5].

**Table 1 pathogens-11-00752-t001:** GPS-HRM 1 primers for identification of *Glaesserella parasuis* using the species-specific gene marker HPS_219690793 and virulence marker *vtaA* for differentiation between virulent and non-virulent *G. parasuis* strains. Primer pair AV1-F and NV1-R targets the leader sequence of *vtaA* (virulence-associated trimeric autotransporter), associated with non-clinical (non-virulent) isolates.

Primer Name	Sequence 5′-3′	Target Gene	Reference	Amplicon Size (bp)	Concentration	Melting Temperature (Tm)
HRM1_GPS_F	AGCTTCCATAAAAGGGAAA	HSP_219690793	this study	96	500 nM	75.8 ± 0.2
HRM1_GPS_R	TGGGAATATCAGACAGGAG		this study		500 nM	
AV1-F vta	AAATATTTAGAGTTATTTGGAGTC	non-virulent *vtaA*	[5]	221	1000 nM	79.65 ± 0.3
NV1-R	CAGAATAAGCAAAATCAGC		[5]		1000 nM	

**Table 2 pathogens-11-00752-t002:** GPS-HRM2 primers for identification of *Glaesserella parasuis* serovars 1, 2, 5, 7, 8, 10, and 15.

Primer Name	Sequence 5′-3′	Serovar	Target Gene	Reference	Amplicon Size (bp)	Concentration	Melting Temperature (Tm)
funB-H1_f	CTGTGTATAATCTATCCCCGATCATCAGC	serovar 1	*funB*	[24]	180	400 nM	77.85 ± 0.15
funB-H1_r	GTCCAACAGAATTTGGACCAATTCCTG	serovar 1		[24]		400 nM	
2funEF	TCTAGAGAAGACGGGATTAGTGG	serovar 2	*funE*	[28]	84	400 nM	75.6 ± 0.15
2funER	CGGAAGCCACGATTCTATTGC	serovar 2		[28]		400 nM	
5wcwKF	CACTGGATAGAGAGTGGCAG	serovar 5	*wcwK*	[28]	139	400 nM	74.4± 0.1
5wcwKR	GGGCAGTTTTTTTCTATAGATCTT	serovar 5		[28]		400 nM	
7funQF	TTAAGGGGGATGTCAGAGCAAG	serovar 7	*funQ*	[28]	229	200 nM	76.5 ± 0.1
7funQR	CCTGGTCATATAATGGCTGCAC	serovar 7		[28]		200 nM	
8scdAF	AAGCAGCAGGTTCTATGGAGTC	serovar 8	*scdA*	[28]	241	200 nM	77.45 ± 0.15
8scdAR	AAAACGCCACGAAATGACATC	serovar 8		[28]		200 nM	
10funXF	AGAGAATTGGGCAAGGCATG	serovar 10	*funX*	[28]	134	300 nM	73.6 ± 0.2
10funXR	CTCGCCCATAAATGTCACCAAC	serovar 10		[28]		300 nM	
15funIF	GGTTTTGTGTGGGGTGGATTTG	serovar 15	*funI*	[28]	113	400 nM	72.7 ± 0.1
15funIR	CATTTGTTGGATGTACGCCATTG	serovar 15		[28]		400 nM	

**Table 3 pathogens-11-00752-t003:** GPS-HRM3 primers for identification of *Glaesserella parasuis* serotypes 3, 4, 6, 9, 11, 12, 13, and 14.

Primer Name	Sequence 5′-3′	Serovar	Target Gene	Reference	Amplicon Size (bp)	Concentration	Melting Temperature (Tm)
3_HRM_F	GTGTTTATCCTGACTTGGCTGTC	serovar 3	*glyC*	this study	129	450 nM	76 ± 0.1
3glyCR	ATCCGCCCAATATGCCTTTC	serovar 3		[28]		450 nM	
4wciPF	ACAGGAGGGGTTGAAAAGACC	serovar 4	*wciP*	[28]	191	200 nM	76.6 ± 0.1
4wciPR	CAAGATTCCCCCAATCATCTGC	serovar 4		[28]		200 nM	
6funLF	TGGAGCGAATCACACTTATCG	serovar 6	*funL*	[28]	122	350 nM	74.4 ± 0.1
6funLR	CCGCTTCCCATACCATACAAC	serovar 6		[28]		350 nM	
9funVF	GGGACTGAAACTGGTTCTGTTC	serovar 9	*funV*	[28]	173	200 nM	77 ± 0.1
9funVR	AATACTCCCCCACCAAAGAACC	serovar 9		[28]		200 nM	
11amtAF	TGGTGCTTGGTCTTTTTGCC	serovar 11	*amtA*	[28]	180	350 nM	79.4 ± 0.1
11amtAR	AAAGAGTCGTGAACCCAACG	serovar 11		[28]		350 nM	
12_HRM_F	ATGAAAATTGATTTCGTACTACCTTGG	serovar 12	*wcwK*	this study	156	250 nM	75.2 ± 0.1
12_HRM_R	AGACCTAAGAACATATCTTAGAGTTCC	serovar 12		this study		250 nM	
13waaLF	GGGGTTTTAGCATTTGTATTCGG	serovar 13	*waaL*	[28]	159	200 nM	73.7 ± 0.15
13waaLR	ATTCGCTCCTTGCTCAACTC	serovar 13		[28]		200 nM	
14funABF	ACCTGCAGGCAATGTAACTC	serovar 14	*funAB*	[28]	271	300 nM	72.4 ± 0.1
14funABR	ACCCATTATCCCCAACCCAAC	serovar 14		[28]		300 nM	

**Table 4 pathogens-11-00752-t004:** Clinical isolates of *Glaesserella parasuis* used in the study.

*G. parasuis* Isolates	Year	Serovar ^a^	Virulence ^b^	Origin	Anamnesis/Clinical Signs
SS626 SK1	2007	1	virulent	lung	unknown
SS626 SK2	2007	7	virulent	lung	unknown
PP396	2016	2	virulent	joint	fever, increased herd mortality
SS3873	2017	2	virulent	joint	neurological dysfunction, fever
SS3875	2017	2	virulent	joint	neurological dysfunction
SS3939	2017	2	virulent	joint	inflammation joints
PP733	2018	2	virulent	brain	sudden death, fever, coughing
PP749	2018	2	virulent	lung	diarrhea, poor condition
PP797	2019	13	virulent	brain	meningitis, neurological dysfunction
PP808	2019	4	virulent	lung	pneumonia
PP849	2020	7	virulent	lung	coughing, dyspnea
SS5061	2020	13	virulent	joint	swollen joints
PP879	2021	14	virulent	lung	coughing
21-640/1	2021	4	virulent	lung	increased herd morbidity
SS5603	2021	4	virulent	joint	polyarthritis, oedema in head
PP903	2021	7	virulent	brain	neurological dysfunction
22-180	2022	7	virulent	brain	lameness
22-676/4	2022	2	virulent	brain	poor condition

^a^ multiplex PCR according to Howell et al., 2015 and Lacouture et al., 2017 [24,31]; ^b^ leader sequence PCR (LS-PCR) according to Galofré-Milà et al., 2017 [5].

**Table 5 pathogens-11-00752-t005:** *Glaesserella parasuis* reference strains used for the development of high-resolution melting (HRM) assays. All reference strains were obtained from Judith Rodhe from the Institute of Microbiology in Hannover, Germany. V, virulent; NV, non-virulent.

Species	Strain	Serovar	LS-PCR ^a^
*G. parasuis*	nr. 4	1	V
*G. parasuis*	SW140	2	V
*G. parasuis*	SW114	3	NV
*G. parasuis*	SW124	4	V
*G. parasuis*	Nagasaki	5	V
*G. parasuis*	131	6	NV
*G. parasuis*	174	7	V
*G. parasuis*	C5	8	NV
*G. parasuis*	D74	9	NV
*G. parasuis*	H555	10	NV
*G. parasuis*	H465	11	V
*G. parasuis*	H425	12	V
*G. parasuis*	84-17975	13	V
*G. parasuis*	84-22113	14	V
*G. parasuis*	84-15995	15	V

^a^ leader sequence PCR (LS-PCR) according to Galofré-Milà et al., 2017 [5].

## Data Availability

The data presented in this study are available in the Appendix A.

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
