# Peer review of "Identification of Glaesserella parasuis and Differentiation of Its 15 Serovars Using High-Resolution Melting Assays"

_pathogens, 2022, doi:10.3390/pathogens11070752_

Round 1

Reviewer 1 Report

Authors intend to develop a combination of HRM1, 2, 3 to differentiate among the 15 serovars of G. parasuis found in pigs. The manuscript is in general well written. 

Abstract: results of 18 clinical isolates (Figure 4) should be included here.  Methods without real application is not meaningful.

Table 1, column 5: can be replaced by one sentence in the note.

line 86: replace "18" with "Eighteen".

Table 2, column 6" replace "bad" condition with "poor" condition.  Replace "pumping respiration" with more specific clinical term, such as tachypnea or dyspnea or something more appropriate as difficult breathing etc.

line 100: I do not see reference 27 cited, nor elsewhere

Table 4 and 5: is there an estimation of "resolution power" regarding Tm, for each HRM? For example a difference of 0.4 centigrade (77.85-77.45) is resolvable. If yes, please mention it for each HRM.

Section 2.3, Analytical specificity: the selection of these bacterial strains include many Actinobacillus spp, which usually will not be included in the differential diagnoses in real practice.  I think the inclusion of these bacterial strains should be based on (and listed by) 3 principles: gram negative cocobacilli (e.g. other Pasturellaceae) like G. parasuis; nasal commensal microorganisms like G. suis; and several bacterial involved in causing similar lesions of GD, like S suis, non-hemolytic E. coli, Mycoplasma hyorhinis, M. hyosynovia, Erysipelothrix rhusiopathiae, Salmonella choleraesuis.

Line 174-175 and 178-179: the statements are redundant, merge them.

Figure 2:  GPS 11 is present in both 2A and 2B, I suggest you discuss this issue.

line 224: please clarify "serovar 15" or "15"? Why specifically choose serovar 15  for serial dilution? Is there any reason? You can discuss this.

Figure 4 A, B, C: GPS11 is present in all of them.  Notice that serovar 11 is not among your 18 clinical isolates.  Is this GPS11 the same as strain H465 of Table 1 and included as some kind of control for validity of HRM? You can discuss this.

Discussion: elaborate more. Lines 264-282 are repeats of your results and introduction.  Only lines 283-295 are real discussion.  You may elaborate more based on the diagnostic or veterinary point of views, like the logics mentioned in choosing bacterial strains for analyzing specificity.

Reviewer 2 Report

I reviewed the manuscript entitled “Identification of Glaesserella parasuis and differentiation of its 15 serovars by high resolution melting assays”. I this study authors report the development and validation of a HRM approach for the diagnosis of GPS.

Overall, I consider it is an interesting study, different sections in the manuscript are well-written. However, I consider that there are some issues that need to be properly addressed by the authors before considering this study for publication.

A)    Introduction section, I would suggest authors including more information regarding the overall epidemiological situation of Glaesserella parasuis around the world. Also, to include more information about the genetic variability of this bacteria (see for example Wan et al., 2020. doi: 10.7717/peerj.9293).

B)    For the primers developed for this study HRM1 GPS_F and HRM1_GPS-R, I suggest providing more details about the genetic identity in these targets among different isolates Glaesserella parasuis. Is it expected to cover all reported isolates?

C)    About the methodology used by authors for the validation of this test, why the performance of HRM1 GPS_F and HRM1_GPS-R, I was not compared with a previously standardized real time test? I think it is important to put in context the values of analytical sensitivity expressed for the test developed herein.

D)    In the results presented in figure 1 and figure 4 A, there are two melting curves produced by the primers AV1-F vta and NV1-R, is it an unspecific reaction? Please could you provide more details in the test.

E)     Improve the discussion, expressing more details about advantages and disadvantages of this test in comparison to previous ones (cost, sensitivity etc.)

Reviewer 3 Report

Identification of Glaesserella parasuis and differentiation of its 15 serovars by high resolution melting assays by Scherrer et al is reviewed.  The manuscript describes development and validation of an elegant HRM assay that enables identification, discrimination of potential virulence and serovars of strains.  Overall the manuscript is well written and provides strong evidence this is a suitable assay for the purpose.  Although many laboratories do not utilize HRM for routine diagnostics, those that have this capability could leverage this assay for robust strain characterization.  The collection of strains used is adequate, however, in the clinical isolate collection there were not non-virulent stains used.  Understandably these are not common in diagnostic submissions, but should be recognized.  There are a few instances of improper italicization and bolding of text that could be revised.  Additionally, the authors use "symptoms" throughout, animals cannot report symptoms so this should be clinical signs.  

Specific Comments: 

Line 100-101- Please provide some details on the in silico analysis of the new primers.  How was this done? Please include some information in results on this analysis. 

Line 103- How was DNA extracted from strains/isolates? Please provide details

Line 114- How was the analysis done and was there any specifics to interpretation?

Figure 3 and Table 6 Could me moved to supplementary information

Line 268-270- What is known about serovar specificity and vaccines?  Does vaccine strategies require inclusion of specific serovars for protection?  That would enhance the impact of this assay if swine producers can match field strains with vaccines to maximize protection.  

Round 2

Reviewer 1 Report

There is significant improvement in the revised version of this  manuscript, in particular the discussion and technical aspect of the assays.  

Reviewer 2 Report

Thank you the authors for your responses, at this point, I don't have more concerns about this study.